# Modified Graphene-FEVE Composite Coatings: Application in the Repair of Ancient Architectural Color Paintings

**Peng Fu [1], Ge-Le Teri [1], Xiao-Lian Chao [1], Jing Li [1], Yu-Hu Li [1,*] and Hong Yang [2,*]**

[1] Engineering Research Center of Historical Cultural Heritage Conservation, Ministry of Education, School of Materials Science and Engineering, Shaanxi Normal University, No. 620, West Chang'an Avenue, Chang'an District, Xi'an 710119, China; fupeng@snnu.edu.cn (P.F.); terigelesnnu@163.com (G.-L.T.); chaoxl@snnu.edu.cn (X.-L.C.); lijing9669@126.com (J.L.)

[2] The Imperial Palace Museum, Beijing 100009, China

* Correspondence: liyuhu@snnu.edu.cn (Y.-H.L.); yanghong960518@163.com (H.Y.)

**Abstract:** In recent years, based on the urgent need in the field of cultural heritage conservation, the research and development of coatings have attracted much attention. FEVE (trifluorovinyl chloride and vinyl ether copolymer) is one of the reinforcing materials in the protective coatings of color paint. However, it has problems such as compactness, low tensile strength, and poor resistance to aging. Therefore, modified graphene was introduced and combined with FEVE coatings (FEVE/m-GO) to optimize their adhesion, compactness, resistance to corrosion, and performance at shielding the paintings from ultraviolet light. The structural features of the hybrid films were characterized by UV–Vis spectroscopy, transmission electron microscopy, infrared spectroscopy, etc. In addition, the water absorption, mechanical properties, color difference test, and aging resistance of the FEVE/m-GO and simulated samples were investigated. The results showed that the hybrid film with 0.04% m-GO incorporation as an effective consolidant exhibited outstanding comprehensive performance. This composite material was used in the protection and consolidation of the Sanyou Xuan ancient architectural color painting in the Palace Museum, which opened up a new way of thinking about the long-term conservation of color paintings.

**Keywords:** cultural heritage; FEVE; modified grapheme; consolidation; color paintings

## 1. Introduction

Ancient architectural color painting is one of the ancient Chinese art forms and an important inheritance of world cultural heritage. Because of the characteristics of the production process of the ancient paintings from the Qing Dynasty [1–3], research on its restoration process has attracted wide attention. Ancient Qing Dynasty paintings are generally divided into three layers: The first layer is wood for support and the second layer is mortar. The mortar consists of Tung oil [4,5], flour [6], brick ash [7], lime water, and other materials combined in different proportions. The third layer is the pigments layer, which consists of glue made from animal products and different pigments that were mixed and painted onto the surface of the mortar. Complex processes undoubtedly increase the difficulty of the repair work. However, most of the ancient architectural color paintings have been exposed to the external environment for a long time [8]. Unfortunately, the synergistic effect of multiple factors, such as light, water, salt, etc., has resulted in a large amount of powdering, peeling, warping, etc. of many of the ancient architectural color paintings dating back to the Qing Dynasty. This has ultimately led to the complete absence of the pigment layer in many instances (Figure 1). Therefore,

it is important to develop protective materials that protect against ultraviolet light, and corrosion, and have anti-fouling properties.

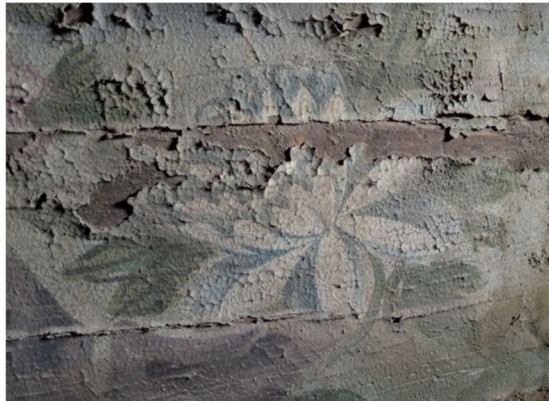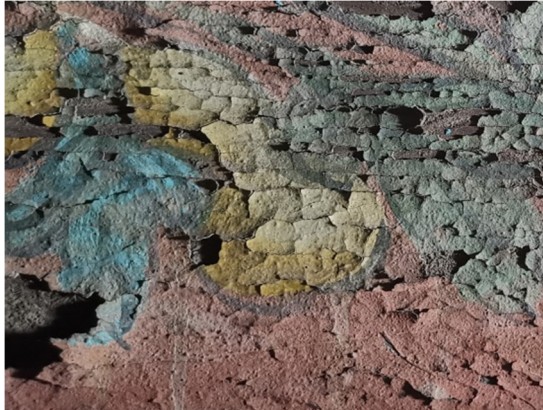

**Figure 1.** The color painting surface of Sanyou Xuan, the Forbidden City. Due to the synergistic effect of light and salt, the surface of color painting appears powdering, peeling, and warping.

At present, acrylic polymers are the organic synthetic materials most often used to reinforce and protect the ancient paintings. Acrylic polymers are widely used for their transparent film-forming ability and good adhesion properties, e.g., a 70/30% mixture of poly(ethyl methacrylate-methyl acrylate) and B72 (Paraloid B72) [9,10]. However, after a long period of practical application, PB72 has demonstrated some shortcomings [11–13], such as a poor resistance to aging. FEVE fluorocarbon resin [14,15] has attracted much attention in the field of research on, and development of, coatings because of its structural characteristics. Based on the chemical structure, the fluoroolefin unit protects the unstable vinyl ether unit from oxidative erosion. Alkenyl ethers (or esters) on the side chains make the resin soluble and transparent. The carboxyl groups ensure that the pigment is wettable and adhesive, and the hydroxyl groups provide the cross-linking groups (Figure 2a) [16,17]. The special structure endows it with good weatherability, antifouling behavior, low VOC emissions, and other advantages. However, some problems remain, such as poor density, low tensile strength, poor adhesion, and poor resistance to aging. The excellent properties of graphene [15,18] include only needing a small amount of doping to significantly improve its light shielding abilities [19], surface energy [20], mechanical strength [21], resistant to corrosion [22,23] and the antimicrobial properties of the matrix resin [24,25]. However, graphene has poor hydrophilicity; consequently, much work has been done to modify graphene to improve its compatibility by combining it with hydrophilic groups [26,27]. However, few relevant studies have been completed in the field of protecting cultural relics, especially in the consolidation and protection of color paintings.

In the present work, modified graphene was combined with a FEVE coating to prepare a composite coating (FEVE/m-GO). First, the graphene was sulfonated and then blended with the FEVE resin in different proportions (F/G-*X*). Compared with the original FEVE, the FEVE/m-GO had better mechanical and optical properties, and resistance to corrosion as determined by tests of the film morphology and performance, and a comparison of the performance of samples. The color difference between B72 and FEVE/m-GO was very small. Compared with B72, the FEVE/m-GO had better resistance to ultraviolet light and salt. The innovation of this paper is that the aging resistance (ultraviolet and salt aging), mechanical properties and hydrophobicity of FEVE-based coatings are improved by doping trace amounts of modified graphene oxide, which is attributed to the high ductility and light shielding properties of graphene. Meanwhile, the effect of graphene oxide modification is to make it disperse uniformly in solvents and FEVE for a long time, which is suitable for the solidification of ancient architectural paintings. The small-scale application of this composite to the Sanyou Xuan color painting (Figure 2b) in the Palace Museum, showed that this composite material has great potential for restoring and protecting color paintings.

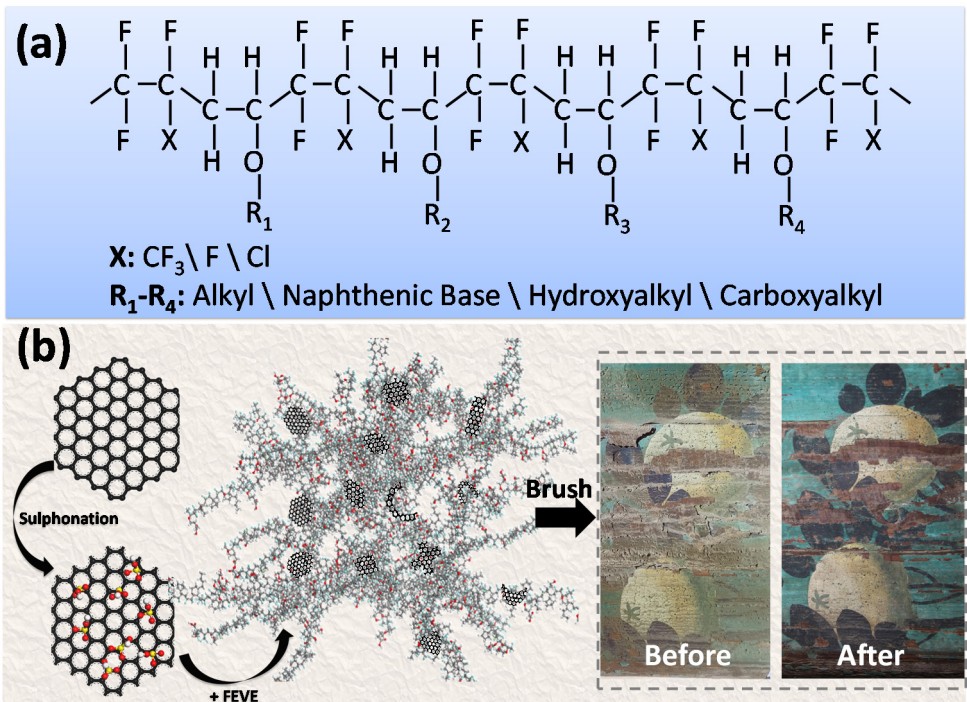

**Figure 2.** (**a**) Molecular formula of FEVE. (**b**) Diagram of the synthesis of FEVE/m-GO and Sanyou Xuan color painting application.

## 2. Materials and Experiment

### 2.1. Chemicals and Apparatus

FEVE was purchased from the Dalian Zhenbang Co., Ltd. (Dalian, China). Graphene oxide (GO) was obtained from the Nanjing Ji Cang Nano Technology Co. Ltd. (Nanjing, China). Paraoid B72 was provided by the Palace Museum and was produced by the Acetone Chemical Reagent Co., Ltd. (Beijing, China). Sodium benzene sulfonate (NASS) was supplied by Konica Minolta Co. (Tokyo, Japan). Potassium persulfate (KPS) was supplied by the Shanghai Lingfeng Chemical Reagent Co., Ltd. (Shanghai, China). The pigments (emerald green, vermilion, and ultramarine) were purchased from Beijing Tianya Pigment Co., Ltd. (Beijing, China). Cooked tung oil, brick ash, flour and lime water were provided by the Beijing Ancient Architecture and Landscape Design Co., Ltd. (Beijing, China).

The QHQ-A pencil hardness tester (App Instrument Co., Ltd., Quzhou, China), complies with ISO 15184 [28]. An Instron 5943 universal material testing machine (Instron Engineering Corporation, Norwood, MA, USA) was used at a test speed of 5 mm/min. A UV-vis spectrophotometer (UV-Vis, PerkinElmer, Lambda 950) capable of measuring across wavelengths ranging from 200 to 3300 nm was used. An infrared spectrometer (Bruker, Tensor27, Leipzig, Germany), and field emission transmission electron microscopy (TEM, FEI, Tecnai G2 F20, Portland, OR, USA) were used. The morphology features of the samples was studied by scanning electron microscopy with energy dispersive X-ray (SEM-EDX, Hitachi High-tech Co. Ltd., SU-8020, Tokyo, Japan). In this contact angle test, a video optical contact angle tester (Dataphysics, OCA20, Stuttgart, Germany) is used to take photos of water droplets, and 2 μL of water was contained in the deposition drop.

### 2.2. Preparation of the Materials

Modification of the GO:GO was placed in an eggplant flask, and initiator KPS, NaSS and water were added sequentially, and then deoxygenated using $N_2$ for 10 min. After reacting in an ultrasonic crusher for 1 h, the dispersion was decanted, centrifuged, and vacuum freeze-dried, and labeled as m-GO.

Preparation of the FEVE/m-GO: Different quantities (0.01%, 0.02%, 0.04%, 0.08%, 0.16%) of m-GO were dissolved in a certain volume of ultrapure water and then mixed with FEVE, and labeled F/G-1, F/G-2, F/G-3, F/G-4, F/G-5, respectively.

B72: B72 powder was dissolved in acetone at a concentration of 5% and labeled as B72.

FEVE: The FEVE coating was diluted with ultrapure water to obtain a 5% aqueous solution and was labeled as FEVE.

### 2.3. Methods Used to Test the Properties of the Film

Film water absorption test: The prepared coatings of B72, FEVE, and F/G-*X* (*X* = 1–5) were applied as a coating onto 76 mm × 26 mm slides to form films, and then dried at room temperature for 24 h. The thickness of each film was approximately 100 μm. A scalpel was used to carefully remove the film from each slide. Each film was weighed to an accuracy of 0.0001 g before and after it was immersed in ultrapure water for 24 h. The following formula was used to calculate the amount of water absorbed by each film [29].

$$\text{water absorption ratio} = \frac{W_\text{t} - W_\text{o}}{W_\text{o}} \times 100\% \tag{1}$$

$W_\text{t}$ is the weight of the film after being immersed in ultrapure water for 24 h, and $W_\text{o}$ is the weight of the film prior to being immersed in the ultrapure water.

Mechanical properties of FEVE/m-GO:A QHQ-A pencil hardness tester was used to test the hardness of the film. An Instron 5942 universal testing machine was used to test the shear strength of the coating. The size of each wood specimen to be bonded was 100 mm × 25 mm × 3 mm. The shear area size was 12.5 ± 0.25 mm × 25 mm, and the test speed was 5 mm/min.

### 2.4. Performance of Coatings Applied to the Samples Simulating Mortar

Rectangular boards (5 cm × 5 cm × 3 cm) were sanded with sandpaper (Figure 3a). The cooked Tung oil, flour, lime water, and brick ash were mixed together according to the ratios of the traditional process used by the painters of the ancient Chinese paintings. The mixture was evenly painted onto the boards and placed at room temperature for 48 h to simulate mortar (Figure 3b). Finally, the pigments and animal glue were evenly mixed and painted onto the surface of the simulated mortar (Figure 3c). After drying, the samples were ready for testing.

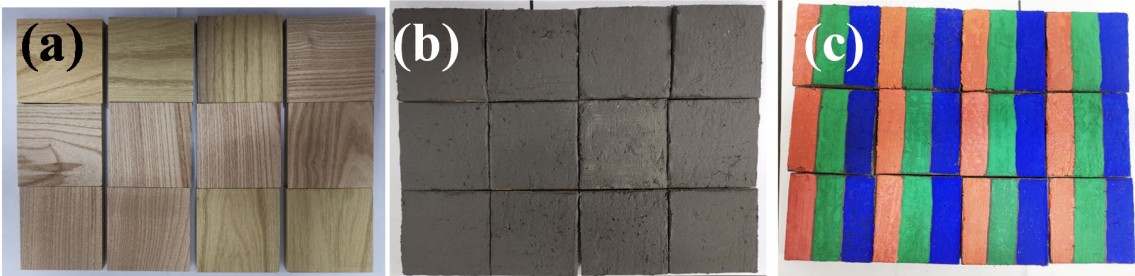

**Figure 3.** (**a**) Wood for support. (**b**) The mortar was applied to the wood. (**c**) Different pigments (vermilion, emerald green, and ultramarine) on the surface of the mortar.

#### 2.4.1. Water Absorption Test of the Samples

Different coatings were applied to the surface of the samples for consolidation. The samples were removed from the constant temperature and humidity box after being placed there for 24 h at *T* = 25 °C and Rh = 50 ± 5%. Each sample surface was placed in contact with a water absorbent stone after the stone had been saturated with water. The water was transported to the surface of the sample by capillary action. The amount of water absorbed by each sample was obtained by using the formula in Section 2.3.

### 2.4.2. Testing the Mechanical Properties of the Samples

The "Scotch tape test" (STT) [21] is a research method to characterize the consolidation efficiency. Transparent tape is cut into a rectangle of the same length and width as the sample (5 cm × 5 cm) and pasted onto the surface of the sample. This is followed by the transparent tape being removed with tweezers at a constant speed. This process was repeated three times at the same speed and the average value was calculated according to the weight loss formula

$$\text{Weight} - \text{loss} = \frac{m_2 - m_1}{a^2} \tag{2}$$

where $m_1$ and $m_2$ represent the weight of the square tape and the weight of the square tape removed from the painting surface, respectively, and $a$ is the length and width of the scotch tape.

### 2.4.3. Aging Resistance Test

Salt Resistance Test

(a) Hygroscopic salt/pigment/consolidant system (hygroscopic salt itself exists in the sample): Vermilion, ultramarine, and emerald green pigments were mixed with 5% hygroscopic salt ($Na_2SO_4$) and a bone glue solution and were then brushed onto the simulated mortar. Finally, the reinforcement coating was applied, and the samples were kept at room temperature for 24 h. All samples were put into a freeze–thaw cycle box, the freeze–thaw aging is set at 24 h cycle and −40 °C–40 °C is divided into 4 stages. Replacement every 6 h, e.g., −40 °C to −20 °C, takes 6 h. This experiment completed a cycle in 24 h, with 60 cycles equal to 60 days. The humidity was maintained at 50%, and the STT test was conducted every 15 days.

(b) Pigment/consolidant/hygroscopic salt system (hygroscopic salt from the outside): Bone glue was mixed with three kinds of pigments, and brushed onto the simulated mortar, and then kept at room temperature for 24 h. Ten milliliters of a 5% sodium sulfate solution was poured into a culture dish, and the absorbent stone was then placed in the culture dish. The strengthened sample pigment layer was placed facing downward onto the absorbent stone and immersed for 3 h. After removal, the sample was put into a freeze–thaw circulation box, with the temperature set to −40 °C–40 °C, humidity at 50%, for 60 days, and the STT test was conducted every 15 days.

UV Aging Test

The samples treated with the coating were placed under a 254MM UV lamp approximately 4–5 cm away from it. The ultraviolet light wavelength and intensity of ultraviolet aging test box are 313 nm and 0.8 W/m$^2$, respectively. Aging lasted for 150 days, and STT tests were performed every 30 days.

Color Difference Test of the Samples

CIE $\Delta L$, $\Delta a$, and $\Delta b$ were used to evaluate the color change of the paint applied to the color painting [30,31].

$$\Delta E = \sqrt{(\Delta L)^2 + (\Delta a)^2 + (\Delta b)^2} \tag{3}$$

where $\Delta E$ is the color difference value, the brightness difference is $\Delta L$, the degrees of difference in red and green is $\Delta a$, and the yellow and blue color difference is $\Delta b$. If the value of $\Delta E$ is large, the color change is large, and vice versa.

## 3. Results and Discussion

### 3.1. Structural Characterization of FEVE/m-GO

The modification of GO can improve its surface structure, and the addition of hydrophilic groups can improve the hydrophilicity of GO. FT-IR spectra of GO and m-GO showed that there were no

obvious absorption peaks in the GO spectra. This was because of the strong absorbance characteristics of carbon materials on the one hand, and because the functional groups on the surface of GO of this raw material were not abundant on the other hand (Figure 4a). In order to improve the dispersion stability of graphene oxide in water, the monomer sodium styrene sulfonate (NaSS) was used as a grafting monomer to carry out free radical polymerization under ultrasound irradiation to generate water-soluble polymer sodium poly(4-styrene sulfonate) (PSS) [32,33]. Meanwhile, the absorption peaks of PSS appeared in the FT-IR spectra of m-GO, among which 1180, 1132, 1047 cm$^{-1}$ were the absorption peaks of sulfonic acid groups [34,35]. This indicated that the monomer NaSS was polymerized and grafted onto the surface of the GO. The effect of GO coated with PSS molecule is obvious, m-GO remains stable in aqueous solution for a long time (about 120 h). However, graphite oxide with the same concentration is poorly dispersed and deposited at the bottom after standing (Supplementary Materials Figure S1). The properties of composite coatings strongly depend on the dispersion of graphene, as shown in Figure 4b. Even after prolonged ultrasonic treatment, the original GO still agglomerated in the FEVE solution and demonstrated extremely poor dispersion. However, the modified m-GO showed a fine flake-like structure and was well dispersed in the emulsion. The results showed that the sulfonation treatment contributed to the stable dispersion of m-GO in the FEVE solution.

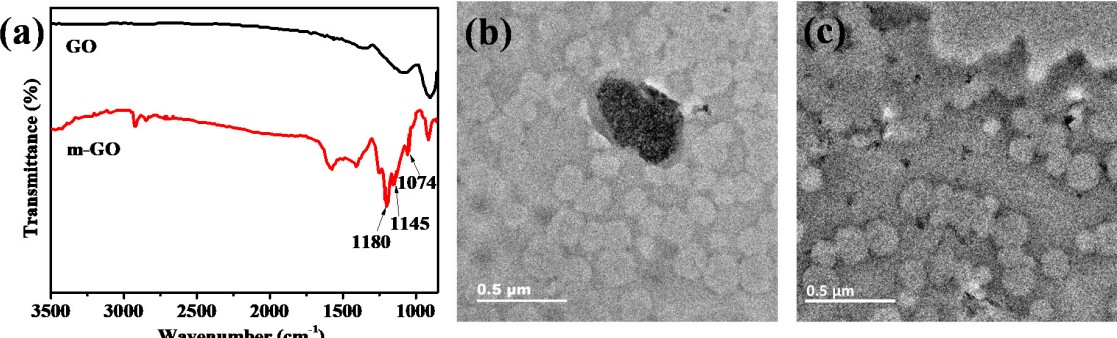

**Figure 4.** (**a**) FT-IR spectra of GO and m-GO. (**b**) TEM of FEVE/GO. (**c**) TEM of FEVE/m-GO.

### 3.2. Properties of the Films

Water absorption is an important index to measure the service range and service life of coatings. If the water absorption rate of the coating is too high, it will cause expansion and affect the surface adhesion of the coating. The coatings of B72, FEVE, and F/G-*X* (*X* = 1–5) were immersed in ultra-pure water for 24 h. The B72 absorbed 32.3% water (Figure 5a), and the water absorption rate of FEVE was 27.4%. However, when m-GO was introduced, the water absorption rate decreased to 14.5%. Simultaneously, the contact angle is another important means of characterizing the surface energy of coatings (the measuring range of contact angle tester is 0–180°). A smaller contact angle indicates that the coating has a higher surface energy and that it will more easily become polluted, i.e., not stay clean. A larger contact angle indicates that the coating has a lower surface energy and that it is more likely to remain clean, i.e., it is less likely to be polluted. This can prevent pollutants such as rainwater and bird droppings from corroding the color of the paintings. After testing, the contact angle of the B72, FEVE, and F/G-5 was 62°, 74°, and 96°, respectively. The contact angle of the F/G-5 was 22° higher than the contact angle of the FEVE (Figure 5a). Compared to FEVE and B72, the FEVE/m-GO was least affected by moisture, which makes it more suitable for applications in humid environments.

It is extremely important to prevent ultraviolet radiation from aging the color of the paintings because of long-term exposure to the external environment. The transmittance and UV resistance of the films were characterized by using a UV–Vis spectrophotometer (Figure 5b,c). The transparency of the films was measured in the 400–700 nm visible light range. To optimize the comparisons of the spectra, a transparency value of 480 nm was selected (Figure 5b). The transmittance of B72, FEVE,

F/G-1, F/G-2, and F/G-3 was 99.06%, 96.47%, 90.82%, 85.55%, and 82.14%, respectively. As these values were all above 80%, they did not affect normal observation of the paintings. The transmittance of F/G-4 and F/G-5 decreased to 75.45% and 67.67%, respectively, with an increase in the concentration of m-GO. A decrease in the transparency of the films was mainly affected by m-GO. At a certain concentration, m-GO can absorb light across a wide range of frequencies. Compared with the UV absorption spectrum (Figure. 5c), B72 has almost no absorption capacity in the range of 300–500 nm, while FEVE has a weak absorption capacity. With the introduction of m-GO, the broad-band absorption peak of F/G-*X* (*X* = 1–5) in the range of 300 to 350 nm indicates that the introduction of m-GO protects the underlying layers against ultraviolet light. In addition, the long-term stability of FEVE/m-GO is an important index to evaluate the practicability of coatings. A series of spectroscopic tests of F/G-3 films after UV aging (wavelength 313 nm) for 120 h was carried out (Figure S2). The FT-IR spectra of F/G-3 maintained its original peak position after aging, and there was no significant change in the UV spectra. It shows that F/G-3 has excellent performance of shielding ultraviolet after long-term ultraviolet radiation.

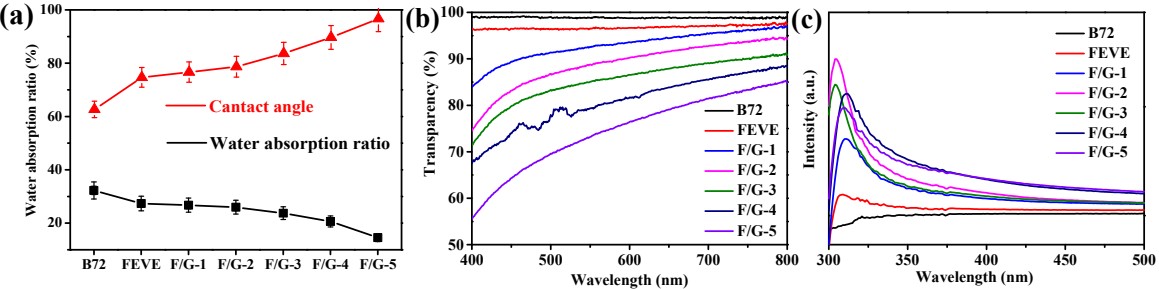

**Figure 5.** (**a**) Water absorption and contact angle of coating film. (**b**) Light transmittance of coating film. (**c**) UV absorption spectrum of coating film.

The pencil hardness test is a rapid method to determine the hardness of film on a smooth surface. It is widely used in the coatings industry and in the protection of cultural relics [36,37]. Using a QHQ-A pencil hardness tester (Table 1), the hardness of B72 and FEVE film was determined to be equivalent to the hardness of an HB and H pencil, respectively. The hardness of the film was positively related to the amount of m-GO added. At an m-GO concentration of 0.16%, the hardness reached the equivalent of a 4H pencil. This was due to the high mechanical strength of graphene, which makes the hybrid film hard. On the other hand, if the hardness is too high, the film will become brittle and the toughness will be reduced [38]. The shear strength of B72 was 4.5291 MPa, which was slightly higher than that of FEVE. However, after 150 days of UV aging, the shear strength of B72 decreased significantly, which indirectly indicated that the resistance to aging of B72 was poor. When the amount of m-GO was 0.04%, the shear strength reached its maximum value of 4.8422 MPa, and it barely changed with aging. However, with an increase in the concentration of m-GO, the shear strength decreased, which indicated that the adhesion of the coating decreased. Combined with the results of the hardness test, this result may be caused by the film becoming more brittle. According to the detection results, the concentration of F/G-3 was more suitable for application.

**Table 1.** Pencil hardness and shear strength of films.

| Sample | Pencil Hardness | Tensile Lap-Shear Strength (MPa) | |
|---|---|---|---|
| | | Before Aging | After Aging |
| B72 | HB | 4.5291 | 3.9168 |
| FEVE | H | 4.3139 | 4.1673 |
| F/G-1 | H | 4.5636 | 4.5796 |
| F/G-2 | 2H | 4.7319 | 4.7382 |
| F/G-3 | 2H | 4.8422 | 4.8514 |
| F/G-4 | 3H | 4.7918 | 4.7619 |
| F/G-5 | 4H | 4.7483 | 4.7361 |

### 3.3. Application of Coatings to the Samples

To verify the practical role of FEVE/m-GO in the restoration of cultural relics, FEVE, B72, and F/G-X were used as consolidation on samples. In the water absorption experiment (Figure 6a), the water absorption of the samples treated with FEVE, and F/G-*X* tended to remain stable and reached their saturation after 20 min, whereas B72 took 40 min to reach saturation. The water absorbed by the FEVE-treated samples after 60 min was 52.33%. The water absorbed by the F/G-1, F/G-2, F/G-3, F/G-4, and F/G-5 samples was 48.24%, 46.17%, 43.24%, 41.49%, and 39.16%, respectively, which was consistent with the water absorption test of the film (Figure 5a). The results show that the combination of m-GO and FEVE can prevent external liquid pollutants from entering the interior of the color painting matrix and causing damage.

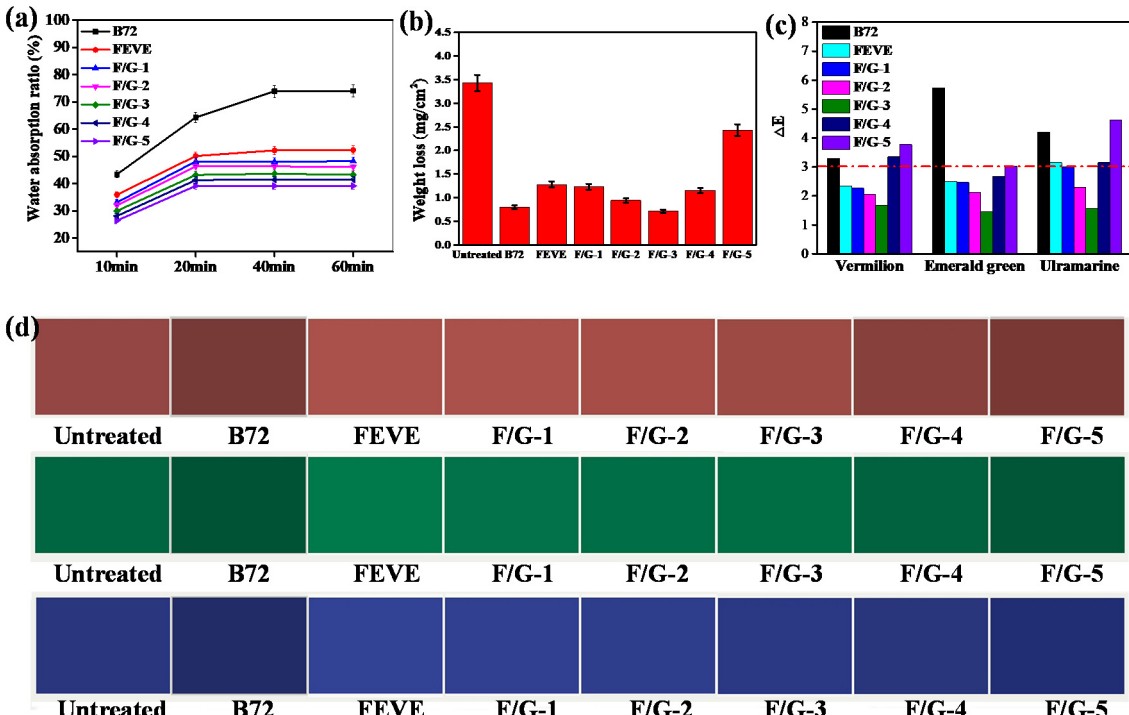

**Figure 6.** (**a**) Capillary water uptake ratio of the simulated samples treated by B72, FEVE, F/G-*X* (*X* = 1–5). (**b**) Weight loss of the simulated samples consolidated with the B72, FEVE, F/G-*X* (*X* = 1–5) coatings in the STT. (**c**) CIELAB ΔE changes of vermilion, ultramarine, and emerald green pigments after consolidation treatment. (**d**) Color change pictures of simulated samples treated with B72, FEVE, F/G-*X* (*X* = 1–5).

The bonding strength of the pigment particles and coatings is an important index used to evaluate the mechanical properties of reinforced color paint. The STT was used to verify the consolidation efficiency of the coatings (Figure 6b). The results showed that several kinds of coatings have obvious effects on the consolidation of color paintings. For the untreated samples, the weight loss was $3.43 \pm 0.5$ mg/cm$^2$, and the weight loss of B72 and FEVE was $0.8 \pm 0.5$ mg/cm$^2$ and $1.28 \pm 0.5$ mg/cm$^2$, respectively. With an increase in the concentration of m-GO, the weight loss first decreased and then increased. The minimum mass loss of F/G-3 was $0.71 \pm 0.5$ mg/cm$^2$. The mass loss of F/G-5 increased to $2.43 \pm 0.5$ mg/cm$^2$, which was consistent with the shear strength of the films (Table 1). The addition of 0.04% m-GO increased the compactness and adhesion of the film, but as the concentration of m-GO increase further, the m-GO exceeded its saturation level inside the FEVE, and this resulted in it accumulating, and the adhesion began to decrease. Consequently, the m-GO concentration of F/G-3 was the most suitable.

In addition, in the field of cultural heritage, it is necessary to minimize the aesthetic changes brought about by the protection and treatment of cultural relics. The CIELAB color space is usually used to define colors. A color difference standard ($\Delta E = 3$) below 3 means a change invisible to the naked eye [39]. As shown in Figure 6c, the color difference of vermilion and ultramarine pigments changed greatly. B72, F/G-4, and F/G-5 exceeded the critical value of 3. The "$\Delta L$" of vermilion, ultramarine, and emerald green pigments after B72 treatment decreased significantly compared with untreated samples, and the color became dark (Figure 6d). Compared with untreated samples, the "$\Delta L$" of the three kinds of pigments after FEVE treatment increased and the color became brighter (Figure 6d). By introducing different concentrations of m-GO and F/G-3, the minimum color changes of the simulated samples were 1.68, 1.46, and 1.55 for vermilion, emerald green, and ultramarine, respectively. Because FEVE has a certain glossiness, the introduction of m-GO increased the coarseness of the film and weakened the glossiness of FEVE, consequently the color difference was reduced. However, if the concentration of m-GO is too high, the color of the picture will become darker. Therefore, the concentration of m-GO should not exceed 0.04%.

More importantly, light and salt are two of the main causes of aging and damage to the pigments of color paintings. UV light not only affects the pigment itself, but also changes the interaction between the pigments and the support/binder. In addition, the chemical corrosion of acid rain, water freezing, and thawing cycles caused by changes in temperature and relative humidity, as well as salt condensation, form sodium sulfate solutions which will permeate the porous matrix on the surface of color painting and cause damage. When the ion concentration is saturated, the salt crystallizes in the porous matrix of the color painting, resulting in a volume expansion of part of the substrate and mechanical stress, which results in the peeling off of the outer surface layer (Figure 1). Therefore, the salt resistance and UV shielding function of the coating are extremely important to the protection of color paintings. The results of the UV aging test are shown in Figure 7a. The weight loss of untreated samples increased exponentially every 30 days, reaching $80.51 \pm 0.5$ mg/cm$^2$ after 150 days. The weight loss of the B72 coatings was within 10mg within 90 days. The weight loss of FEVE was $17.64 \pm 0.5$ mg/cm$^2$ after 150 days. It should be noted that the weight loss of F/G-1, F/G-2, and F/G-3 continued to increase with an increase in illumination time, but the increasing trend rapidly decreased. The weight loss of F/G-4 and F/G-5 did not change with a change in the illumination time and remained unchanged. Compared with Figure 6b, the weight loss of F/G-4 and F/G-5 was consistent with that after UV aging, which was about $2.5 \pm 0.5$ mg/cm$^2$ and $2.9 \pm 0.5$ mg/cm$^2$, respectively. This indicated that the introduction of m-GO had little effect on the UV light of the F/G-4 and F/G-5 samples, but when the content of m-GO was large, it caused a color distortion. In order to explore the ultraviolet resistance of the simulated samples with FEVE/m-GO curing, their morphologies before and after ultraviolet aging were characterized. It can be seen from SEM images and EDX (Figure S3 and Table S1) that there are no obvious changes in morphological characteristics and composition. It further emphasizes the advantages of F/G-3 hybrid film as a protective agent for ancient color paintings.

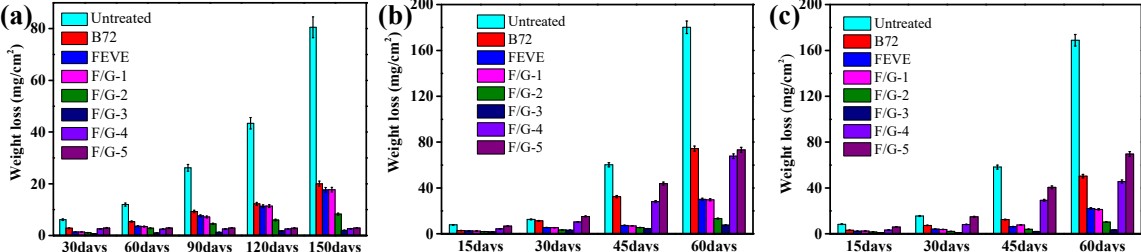

**Figure 7.** (**a**) Weight loss rate of simulated samples after ultraviolet light aging after treatment with B72, FEVE and F/G-*X* (*X* = 1–5). (**b**) Weight loss of simulated samples of hygroscopic salt/pigment/consolidant agent system after B72, FEVE, F/G-*X* (*X* = 1–5) treatment. (**c**) Weight loss of simulated samples of pigment/consolidant/hygroscopic salt system after B72, FEVE, F/G-*X* (*X* = 1–5) treatment.

Subsequently, we studied the hygroscopic salt/pigment/consolidant system of the samples (Figure 7b). The coating on the untreated samples began to separate in large areas after 30 days under the action of 5% sodium sulfate of hygroscopic salt and were loose and shapeless at 60 days. After treatment with a B72 coating, the weight loss at 45 and 60 days was $32.51 \pm 0.5$ mg/cm$^2$ and $74.38 \pm 0.5$ mg/cm$^2$, respectively, which was twice that of FEVE: $30.15 \pm 0.5$ mg/cm$^2$. With the introduction of different concentrations of m-GO, the weight loss of F/G-3 was only $7.62 \pm 0.5$ mg/cm$^2$, which was about eight times less than that of B72, and that of FEVE was about four times less than that of B72. It showed that the introduction of m-GO increased the salt resistance of the samples of color paintings, but when the m-GO concentration continued to increase, due to saturation of the m-GO in FEVE and accumulation, the adhesion decreased, and consequently the weight loss increased. Simultaneously, the experimental results of the pigment/consolidant/hygroscopic salt system were similar to those of the hygroscopic salt/pigment/consolidant system, F/G-3 had the most significant effect on salt resistance, while B72 demonstrated poor salt resistance. By comparing the two experimental methods (Figure 7b,c), the weight loss of the hygroscopic salt/pigment/consolidant system was significantly greater than that of the pigment/consolidant/hygroscopic salt, which resulted in a new understanding of the protection of color paintings. In areas where humidity changes greatly, desalting treatment should be carried out before protecting the paintings with a coating because it will greatly improve the durability of the coating.

## 4. Conclusions

To summarize, we demonstrated an unsophisticated and economical water-based method which combines m-GO with FEVE, as a new type of reinforcement coating for color paintings. The film performance test showed that the contact angle, water absorption, shear strength, and light shielding performance of an F/G-3 film with 0.04% m-GO are significantly improved compared with FEVE and B72. When it was applied to the simulated samples, those treated with F/G-3 absorbed less water and the color difference was acceptable. Compared with the simulated samples of B72 and FEVE, the samples treated by F/G-3 had excellent resistance to aging, and included resistance to salt, and UV light shielding performance. The application of F/G-3 in the consolidation of some color paintings in the Sanyou Xuan of the Forbidden City had a remarkable effect and provided a new idea for research on and development of protective coatings for ancient architectural paintings.

**Supplementary Materials:** The following are available online at http://www.mdpi.com/2079-6412/10/12/1162/s1, Figure S1: Images of water dispersion (0.001 g/mL) of m-GO and GO after 120 h, Figure S2: UV spectra and FT-IR spectra of FEVE/m-GO films before and after aging, Figure S3: SEM of samples before and after aging experiments, Table S1: EDX of samples before and after aging experiments.

**Author Contributions:** Conceptualization, P.F. and Y.-H.L.; data curation, P.F. and G.-L.T.; methodology, X.-L.C.; resources, H.Y.; validation, J.L. All authors have read and agreed to the published version of the manuscript.

**Funding:** This research was sponsored by the Fundamental Research Funds for the Central Universities (2019TS002).

**Acknowledgments:** Thanks to Yang Hong of the Palace Museum for providing a small-scale application pilot for Sanyou Xuan color painting, and thanks to Yu-Hu Li for setting up the framework of the entire project. Thanks for the fund support (the Fundamental Research Funds for the Central Universities, 2019TS002).

**Conflicts of Interest:** The authors declare no conflict of interest.

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
