# Peer review of "Modified Graphene-FEVE Composite Coatings: Application in the Repair of Ancient Architectural Color Paintings"

_coatings, doi:10.3390/coatings10121162_

Round 1
Reviewer 1 Report
It is a very good study with overall adequate presentation of experimental results. Some additions are needed:
1) Authors should further emphasize on the novelty of their work.
2) Some minor typos, grammar and syntax errors should be carefully revised and corrected accordingly.
3) Reference can be even more updated (more recent relative works).
Reviewer 2 Report
The present paper is interesting and well written but it lacks in many aspects.
-Abstract must be rewritten with some of the most important results obtained
-The authors reported that: “With the introduction of m-GO, the broad-band absorption peak of F/G-X (X=1-5) in the range of 300-350nm indicates that the introduction of m-GO has anti ultraviolet performance.” The authors must report the UV-vis spectra of the samples after UV exposure during time. It’s important to understand if the UV absorption features of the m-GO molecules effectively improve the resistance to the UV exposure.
-The authors reported that: “Combined with the result of the hardness test, this result may be caused by the film becoming more brittle. According to the detection results, the concentration of F/G-3 was more suitable.” The authors must explain why this type of sample is more suitable for the applications. How does the specific composition affect the enhancement of the hardness of the material?
-The authors reported that: “However, the absorption peaks of PSS appeared in the m-GO spectra, and 1180, 1132, 1047 cm-1 were the absorption peaks of the sulfonic acid groups [28, 29]. This indicated that the monomer NaSS was polymerized and grafted onto the surface of the GO.”
-The authors must explain why the sulfonic acid groups can influence the innovative characteristics of these coatings. Moreover, FT-IR spectra of the coatings after aging is necessary to be added.
-SEM-EDX images of the samples as prepared and after aging need to be added in order to understand the changes in the morphological features of these materials.
-Error bars must be added for all the figures.
Major revisions are required
Round 2
Reviewer 1 Report
All my comments of the initial submission have been correctly replied and included in the revised manuscript. The quality of this work has been drastically improved after revision and therefore I recommend its publication as it is.
Author Response
Your help was very much appreciated.
Reviewer 2 Report
The paper can be now accepted for the publication since the authors have correctly replied to all the Referee's questions
Author Response
Your help was very much appreciated.